# BEYOND GREEDY RANKING: SLATE OPTIMIZATION VIA LIST-CVAE

**Ray Jiang**[*]   **Sven Gowal**[*]   **Yuqiu Qian**[†]   **Timothy A. Mann**[*]   **Danilo J. Rezende**[*]

## ABSTRACT

The conventional solution to the recommendation problem greedily ranks individual document candidates by prediction scores. However, this method fails to optimize the slate as a whole, and hence, often struggles to capture biases caused by the page layout and document interdepedencies. The slate recommendation problem aims to directly find the optimally ordered subset of documents (*i.e. slates*) that best serve users' interests. Solving this problem is hard due to the combinatorial explosion in all combinations of document candidates and their display positions on the page. Therefore we propose a paradigm shift from the traditional viewpoint of solving a ranking problem to a direct slate generation framework. In this paper, we introduce *List Conditional Variational Auto-Encoders (List-CVAE)*, which learns the joint distribution of documents on the slate conditioned on user responses, and directly generates full slates. Experiments on simulated and real-world data show that List-CVAE outperforms popular comparable ranking methods consistently on various scales of documents corpora.

## 1 INTRODUCTION

Recommender systems modeling is an important machine learning area in the IT industry, powering online advertisement, social networks and various content recommendation services (Schafer et al., 2001; Lu et al., 2015). In the context of document recommendation, its aim is to generate and display an ordered list of "documents" to users (called a "slate" in Swaminathan et al. (2017); Sunehag et al. (2015)), based on both user preferences and documents content. For large scale recommender systems, a common scalable approach at inference time is to first select a small subset of candidate documents $\mathcal{S}$ out of the entire document pool $\mathcal{D}$. This step is called "candidate generation". Then a function approximator such as a neural network (e.g., a Multi-Layer Perceptron (MLP)) called the "ranking model" is used to predict probabilities of user engagements for each document in the small subset $\mathcal{S}$ and greedily generates a slate by sorting the top documents from $\mathcal{S}$ based on estimated prediction scores (Covington et al., 2016). This two-step process is widely popular to solve large scale recommendation problems due to its scalability and fast inference at serving time. The candidate generation step can decrease the number of candidates from millions to hundreds or less, effectively dealing with scalability when faced with a large corpus of documents $\mathcal{D}$. Since $|\mathcal{S}|$ is much smaller than $|\mathcal{D}|$, the ranking model can be reasonably complicated without increasing latency.

However, there are two main problems with this approach. First the candidate generation and the ranking models are not trained jointly, which can lead to having candidates in $\mathcal{S}$ that are not the highest scoring documents of the ranking model. Second and most importantly, the greedy ranking method suffers from numerous biases that come with the visual presentation of the slate and context in which documents are presented, both at training and serving time. For example, there exists positional biases caused by users paying more attention to prominent slate positions (Joachims et al., 2005), and contextual biases, due to interactions between documents presented together in the same slate, such as competition and complementarity, relative attractiveness, etc. (Yue et al., 2010).

In this paper, we propose a paradigm shift from the traditional viewpoint of solving a ranking problem to a direct slate generation framework. We consider a slate "optimal" when it maximizes some type of user engagement feedback, a typical desired scenario in recommender systems. For example, given

---

[*]Google DeepMind, London, UK. Correspondence to: Ray Jiang <rayjiang@google.com>.
[†]The University of Hong Kong

a database of song tracks, the optimal slate can be an ordered list (in time or space) of $k$ songs such that the user ideally likes every song in that list. Another example considers news articles, the optimal slate has $k$ ordered articles such that every article is read by the user. In general, optimality can be defined as a desired user response vector on the slate and the proposed model should be agnostic to these problem-specific definitions. Solving the slate recommendation problem by direct slate generation differs from ranking in that first, the entire slate is used as a training example instead of single documents, preserving numerous biases encoded into the slate that might influence user responses. Secondly, it does not assume that more relevant documents should necessarily be put in earlier positions in the slate at serving time. Our model directly generates slates, taking into account all the relevant biases learned through training.

In this paper, we apply *Conditional Variational Auto-Encoders (CVAEs)* (Kingma et al., 2014; Kingma and Welling, 2013; Jimenez Rezende et al., 2014) to model the distributions of all documents in the same slate conditioned on the user response. All documents in a slate along with their positional, contextual biases are jointly encoded into the latent space, which is then sampled and combined with desired conditioning for direct slate generation, i.e. sampling from the learned conditional joint distribution. Therefore, the model first learns which slates give which type of responses and then directly generates similar slates given a desired response vector as the conditioning at inference time. We call our proposed model **List-CVAE**. The **key contributions** of our work are:

1. To the best of our knowledge, this is the first model that provides a conditional generative modeling framework for slate recommendation by direct generation. It does not necessarily require a candidate generator at inference time and is flexible enough to work with any visual presentation of the slate as long as the ordering of display positions is fixed throughout training and inference times.

2. To deal with the problem at scale, we introduce an architecture that uses pretrained document embeddings combined with a negatively downsampled $k$-head softmax layer within the List-CVAE model, where $k$ is the slate size.

The structure of this paper is the following. First we introduce related work using various CVAE-type models as well as other approaches to solve the slate generation problem. Next we introduce our List-CVAE modeling approach. The last part of the paper is devoted to experiments on both simulated and the real-world datasets.

## 2 RELATED WORK

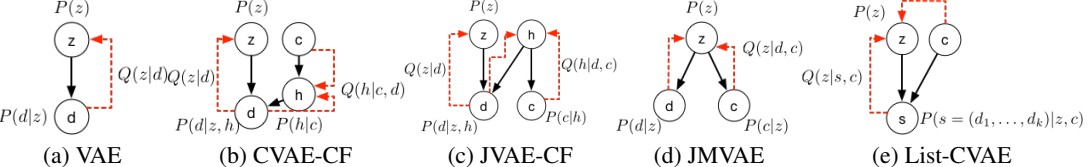

    (a) VAE         (b) CVAE-CF        (c) JVAE-CF        (d) JMVAE        (e) List-CVAE

Figure 1: Comparison of related variants of VAE models. Note that user variables are not included in the graphs for clarity. (a) VAE; (b) CVAE-CF with auxiliary variables; (c) Joint Variational Auto-Encoder-Collaborative Filtering (JVAE-CF); (d) JMVAE; and, (e) List-CVAE (ours) with the whole slate as input.

Traditional matrix factorization techniques have been applied to recommender systems with success in modeling competitions such as the Netflix Prize (Koren et al., 2009). Later research emerged on using autoencoders to improve on the results of matrix factorization (Wu et al., 2016; Wang et al., 2015) (CDAE, CDL). More recently several works use Boltzmann Machines (Abdollahi and Nasraoui, 2016) and variants of VAE models in the Collaborative Filtering (CF) paradigm to model recommender systems (Li and She, 2017; Lee et al., 2017; Liang et al., 2018) (Collaborative VAE, JMVAE, CVAE-CF, JVAE-CF). See Figure 1 for model structure comparisons. In this paper, unless specified otherwise, the user features and any context are routinely considered part of the conditioning variables (in Appendix A Personalization Test, we test List-CVAE generating personalized slates for

different users). These models have primarily focused on modeling individual document or pairs of documents in the slate and applying greedy ordering at inference time.

Our model is also using a VAE type structure and in particular, is closely related to the Joint Multimodel Variational Auto-Encoder (JMVAE) architecture (Figure 1d). However, we use whole slates as input instead of single documents, and directly generate slates instead of using greedy ranking by prediction scores.

Other relevant work from the Information Retrieval (IR) literature are listwise ranking methods (Cao et al., 2007; Xia et al., 2008; Shi et al., 2010; Huang et al., 2015; Ai et al., 2018). These methods use listwise loss functions that take the contexts and positions of training examples into account. However, they eventually assign a prediction score for each document and greedily rank them at inference time.

In the Reinforcement Learning (RL) literature, Sunehag et al. (2015) view the whole slates as actions and use a deterministic policy gradient update to learn a policy that generates these actions, given concatenated document features as input.

Finally, the framework proposed by (Wang et al., 2016) predicts user engagement for document and position pairs. It optimizes whole page layouts at inference time but may suffer from poor scalability due to the combinatorial explosion of all possible document position pairs.

# 3 METHOD

## 3.1 PROBLEM SETUP

We formally define the slate recommendation problem as follows. Let $\mathcal{D}$ denote a corpus of documents and let $k$ be the slate size. Then let $\mathbf{r} = (r_1, \ldots, r_k)$ be the user response vector, where $r_i \in \mathcal{R}$ is the user's response on document $d_i$. For example, if the problem is to maximize the number of clicks on a slate, then let $r_i \in \{0, 1\}$ denote whether the document $d_i$ is clicked, and thus an optimal slate $\mathbf{s} = (d_1, d_2, \ldots, d_k)$ where $d_i \in \mathcal{D}$ is such that $\mathbf{s}$ maximizes $\mathbb{E}[\sum_{i=1}^{k} r_i]$.

## 3.2 VARIATIONAL AUTO-ENCODERS

Variational Auto-Encoders (VAEs) are latent-variable models that define a joint density $P_\theta(x, z)$ between observed variables $x$ and latent variables $z$ parametrized by a vector $\theta$. Training such models requires marginalizing the latent variables in order to maximize the data likelihood $P_\theta(x) = \int P_\theta(x, z) dz$. Since we cannot solve this marginalization explicitly, we resort to a variational approximation. For this, a variational posterior density $Q_\phi(z|x)$ parametrized by a vector $\phi$ is introduced and we optimize the variational Evidence Lower-Bound (ELBO) on the data log-likelihood:

$$\log P_\theta(x) = \mathrm{KL}\left[Q_\phi(z|x) \| P_\theta(z|x) + \mathbb{E}_{Q_\phi(z|x)}\left[-\log Q_\phi(z|x) + \log P_\theta(x, z)\right], \quad (1)$$
$$\geq -\mathrm{KL}\left[Q_\phi(z|x) \| P_\theta(z)\right] + \mathbb{E}_{Q_\phi(z|x)}\left[\log P_\theta(x|z)\right], \quad (2)$$

where KL is the Kullback–Leibler divergence and where $P_\theta(z)$ is a prior distribution over latent variables. In a Conditional VAE (CVAE) we extend the distributions $P_\theta(x, z)$ and $Q_\phi(z|x)$ to also depend on an external condition $c$. The corresponding distributions are indicated by $P_\theta(x, z|c)$ and $Q_\phi(z|x, c)$. Taking the conditioning $c$ into account, we can write the variational loss to minimize as

$$\mathcal{L}_{\mathrm{CVAE}} = \mathrm{KL}\left[Q_\phi(z|x, c) \| P_\theta(z|c)\right] - \mathbb{E}_{Q_\phi(z|x, c)}\left[\log P_\theta(x|z, c)\right]. \quad (3)$$

## 3.3 OUR MODEL

We assume that the slates $\mathbf{s} = (d_1, d_2, \ldots d_k)$ and the user response vectors $\mathbf{r}$ are jointly drawn from a distribution $\mathbb{P}_{\mathcal{D}^k \times \mathcal{R}^k}$. In this paper, we use a CVAE to model the joint distribution of all

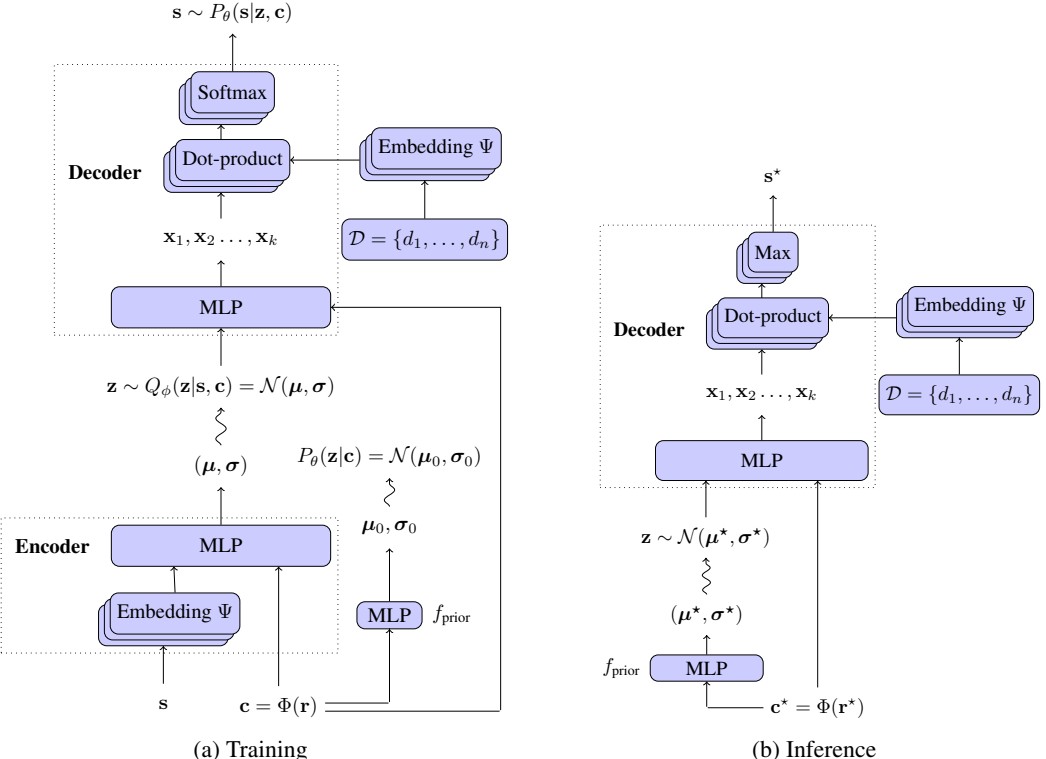

(a) Training                                    (b) Inference

Figure 2: Structure of List-CVAE for both (a) training and (b) inference. $\mathbf{s} = (d_1, d_2, \ldots, d_k)$ is the input slate. $\mathbf{c} = \Phi(\mathbf{r})$ is the conditioning vector, where $\mathbf{r} = (r_1, r_2, \ldots, r_k)$ is the user responses on the slate $\mathbf{s}$. The concatenation of $\mathbf{s}$ and $\mathbf{c}$ makes the input vector to the encoder. The latent variable $\mathbf{z} \in \mathbb{R}^m$ has a learned prior distribution $\mathcal{N}(\boldsymbol{\mu}_0, \boldsymbol{\sigma}_0)$. The raw output from the decoder are $k$ vectors $\mathbf{x}_1, \mathbf{x}_2 \ldots, \mathbf{x}_k$, each of which is mapped to a real document through taking the dot product with the matrix $\Phi$ containing all document embeddings. Thus produced $k$ vectors of logits are then passed to the negatively downsampled $k$-head softmax operation. At inference time, $\mathbf{c}^{\star}$ is the ideal condition whose concatenation with sampled $\mathbf{z}$ is the input to the decoder.

documents in the slate conditioned on the user responses $\mathbf{r}$, i.e. $\mathbb{P}(d_1, d_2, \ldots d_k | \mathbf{r})$. At inference time, the List-CVAE model attempts to generate an optimal slate by conditioning on the ideal user response $\mathbf{r}^{\star}$.

As we explained in Section 1, "optimality" of a slate depends on the task. With that in mind, we define the mapping $\Phi : \mathcal{R}^k \mapsto \mathcal{C}$. It transforms a user response vector $\mathbf{r}$ into a vector in the conditioning space $\mathcal{C}$ that encodes the user engagement metric we wish to optimize for. For instance, if we want to maximize clicks on the slate, we can use the binary click response vectors and set the conditioning to $\mathbf{c} = \Phi(\mathbf{r}) := \sum_{i=0}^{k} r_i$. Then at inference time, the corresponding ideal user response $\mathbf{r}^{\star}$ would be $(1, 1, \ldots, 1)$, and correspondingly the ideal conditioning would be $\mathbf{c}^{\star} = \Phi(\mathbf{r}^{\star}) = \sum_{i=0}^{k} 1 = k$.

As usual with CVAEs, the decoder models a distribution $P_\theta(\mathbf{s}|\mathbf{z}, \mathbf{c})$ that, conditioned on z, is easy to represent. In our case, $P_\theta(\mathbf{s}|\mathbf{z}, \mathbf{c})$ models an independent probability for each document on the slate, represented by a softmax distribution. Note that the documents are only independent to each other conditional on $\mathbf{z}$. In fact, the *marginalized* posterior $P_\theta(\mathbf{s}|\mathbf{c}) = \int_z P_\theta(\mathbf{s}|\mathbf{z}, \mathbf{c}) P_\theta(\mathbf{z}|\mathbf{c}) d\mathbf{z}$ can be arbitrarily complex. When the encoder encodes the input slate $\mathbf{s}$ into the latent space, it learns the joint distribution of the $k$ documents in a fixed order, and thus also encodes any contextual, positional biases between documents and their respective positions into the latent variable $\mathbf{z}$. The decoder learns these biases through reconstruction of the input slates from latent variables $\mathbf{z}$ with conditions. At inference time, the decoder reproduces the input slate distribution from the latent variable $\mathbf{z}$ with the ideal conditioning, taking into account all the biases learned during training time.

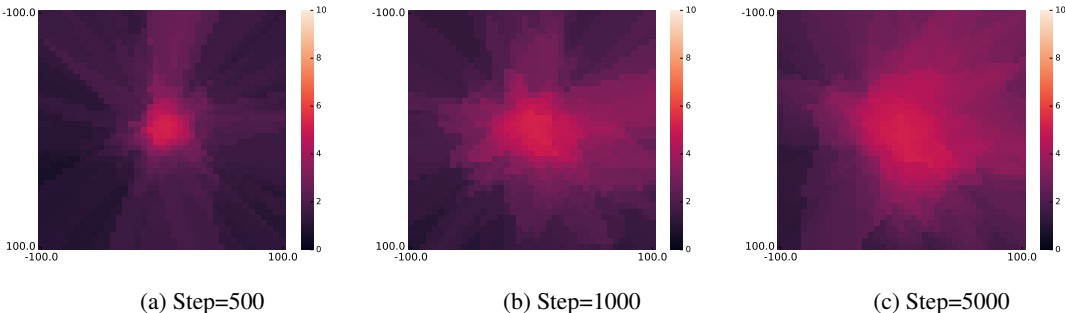

| (a) Step=500 | (b) Step=1000 | (c) Step=5000 |

Figure 3: Predictive prior distribution of the latent variable $z$ in $\mathbb{R}^2$, conditioned on ideal user response $\mathbf{c}^\star = (1, 1, \ldots, 1)$. The color map corresponds to the expected total responses of the corresponding slates. Plots are generated from the simulation experiment with 1000 documents and slate size 10.

To shed light onto what is encoded in the latent space, we simplify the prior distribution of $z$ to be a fixed Gaussian distribution $\mathcal{N}(\mathbf{0}, \mathbf{I})$ in $\mathbb{R}^2$. We train List-CVAE and plot the predictive prior $\mathbf{z}$. As training evolves, generated output slates with low total responses are pushed towards the edge of the latent space while high response slates cluster towards a growing center area (Figure 3). Therefore after training, if we sample $z$ from its prior distribution $\mathcal{N}(\mathbf{0}, \mathbf{I})$ and generate the corresponding output slates, they are more likely to have high total responses.

Since the number of documents in $\mathcal{D}$ can be large, we first embed the documents into a low dimensional space. Let $\Psi : \mathcal{D} \mapsto \mathbb{S}^{q-1}$ be that normalized embedding where $\mathbb{S}^{q-1}$ denotes the unit sphere in $\mathbb{R}^q$. $\Psi$ can easily be pretrained using a standard supervised model that predicts user responses from documents or through a standard auto-encoder technique. For the $i$-th document in the slate, our model produces a vector $x_i$ in $\mathbb{R}^q$ that is then matched to each document from $\mathcal{D}$ *via* a dot-product. This operation produces $k$ vectors of logits for $k$ softmaxes, i.e. the $k$-head softmax. At training time, for large document corpora $\mathcal{D}$, we uniformly randomly downsample negative documents and compute only a small subset of the logits for every training example, therefore efficiently scaling the nearest neighbor search to millions of documents with minimal model quality loss.

We train this model as a CVAE by minimizing the sum of the reconstruction loss and the KL-divergence term:

$$\mathcal{L} = \beta \text{KL}\left[Q_\phi(\mathbf{z} \,|\, \mathbf{s}, \mathbf{c}) \| P_\theta(\mathbf{z} \,|\, \mathbf{c})\right] - \mathbb{E}_{Q_\phi(\mathbf{z}|\mathbf{s},\mathbf{c})}\left[\log P_\theta(\mathbf{s} \,|\, \mathbf{z}, \mathbf{c})\right], \tag{4}$$

where $\beta$ is a function of the training step (Higgins et al., 2017).

During inference, output slates are generated by first sampling $\mathbf{z}$ from the conditionally learned prior distribution $\mathcal{N}(\mu^\star, \sigma^\star)$, concatenating with the ideal condition $\mathbf{c}^\star = \Phi(\mathbf{r}^\star)$, and passed into the decoder, generating $(\mathbf{x}_1, \ldots, \mathbf{x}_k)$ from the learned $P_\theta(\mathbf{s}|\mathbf{z}, \mathbf{c}^\star)$, and finally taking $\arg\max$ over the dot-products with the full embedding matrix independently for each position $i = 1, \ldots, k$.

## 4 EXPERIMENTS

### 4.1 SIMULATION DATA

**Setup:** The simulator generates a random matrix $W \sim \mathcal{N}(\mu, \sigma)^{k \times n \times k \times n}$ where each element $W_{i,d_i,j,d_j}$ represents the interaction between document $d_i$ at position $i$ and document $d_j$ at position $j$, and $n = |\mathcal{D}|$. It simulates biases caused by layouts of documents on the slate (below, we set $\mu = 1$ and $\sigma = 0.5$). Every document $d_i \in \mathcal{D}$ has a probability of engagement $A_i \sim \mathcal{U}([0, 1])$ representing its innate attractiveness. User responses are computed by multiplying $A_i$ with interaction multipliers $W(i, d_i, j, d_j)$ for each document presented before $d_i$ on the slate. Thus the user response

$$r_i \sim \mathcal{B}\left[\left(A_i \prod_{j=1}^{i} W_{i,d_i,j,d_j}\right)\Big|_{[0,1]}\right] \tag{5}$$

for $i = 1, \ldots, k$, where $\mathcal{B}$ represents the Bernoulli distribution.

During training, all models see uniformly randomly generated slates $\mathbf{s} \sim \mathcal{U}(\{1, n\}^k)$ and their generated responses $\mathbf{r}$. During inference time, we generate slates $\mathbf{s}$ by conditioning on $\mathbf{c}^\star = (1, \ldots, 1)$. We do not require document de-duplication since repetition may be desired in certain applications (e.g. generating temporal slates in an online advertisement session). Moreover List-CVAE should learn to produce the optimal slates whether those slates contain duplication or not from learning the training data distribution.

**Evaluation:** For evaluation, we cannot use offline ranking evaluation metrics such as Normalized Discounted Cumulative Gain (NDCG) (Järvelin and Kekäläinen, 2000), Mean Average Precision (MAP) (Baeza-Yates and Ribeiro-Neto, 1999) or Inverse Propensity Score (IPS) (Little and Rubin, 2002), etc. These metrics either require prediction scores for individual documents or assume that more relevant documents should appear in earlier ranking positions, unfairly favoring greedy ranking methods. Moreover, we find it limiting to use various diversity metrics since it is not always the case that a higher diversity-inclusive score gives better slates measured by user's total responses. Even though these metrics may be more transparent, they do not measure our end goal, which is to maximize the expected number of total responses on the generated slates.

Instead, we evaluate the expected number of clicks over the distribution of generated slates and over the distribution of clicks on each document:

$$\mathbb{E}[\sum_{i=1}^{k} r_i] = \sum_{\mathbf{s} \in \{1, \ldots, n\}^k} \mathbb{E}[\sum_{i=1}^{k} r_i | \mathbf{s}] P(\mathbf{s}) = \sum_{\mathbf{s} \in \mathcal{D}^k} \sum_{\mathbf{r} \in \mathcal{R}^k} \sum_{i=1}^{k} r_i P(\mathbf{r}) P(\mathbf{s}). \tag{6}$$

In practice, we distill the simulated environment of Eq. 5 using the cross-entropy loss onto a neural network model that officiates as our new simulation environment. The model consists of an embedding layer, which encodes documents into 8-dimensional embeddings. It then concatenates the embeddings of all the documents that form a slate and follows this concatenation with two hidden layers and a final `softmax` layer that predicts the slate response amongst the $2^k$ possible responses. Thus we call it the "response model". We use the response model to predict user responses on 100,000 sampled output slates for evaluation purposes. This allows us to accurately evaluate our output slates by List-CVAE and all other baseline models.

**Baselines:** Our experiments compare List-CVAE with several greedy ranking baselines that are often deployed in industry productions, namely Greedy MLP, Pairwise MLP, Position MLP and Greedy Long Short-Term Memory (LSTM) models. In addition to the greedy baselines, we also compare against auto-regressive (AR) versions of Position MLP and LSTM, as well as randomly-selected slates from the training set as a sanity check. **List-CVAE** generates slates $\mathbf{s} = \arg\max_{s \in \{1, \ldots, n\}^k} P_\theta(\mathbf{s} | \mathbf{z}, \mathbf{c}^\star)$. The encoder and decoder of List-CVAE, as well as all the MLP-type models consist of two fully-connected neural network layers of the same size. **Greedy MLP** trains on $(d_i, r_i)$ pairs and outputs the greedy slate consisting of the top $k$ highest $\hat{P}(r|d)$ scoring documents. **Pairwise MLP** is an MLP model with a pairwise ranking loss $\mathcal{L} = \alpha \mathcal{L}_x + (1 - \alpha) \mathcal{L}(\hat{P}(x_1) - \hat{P}(x_2) + \eta)$ where $\mathcal{L}_x$ is the cross entropy loss and $(x_1, x_2)$ are pairs of documents randomly selected with different user responses from the same slate. We sweep on hyperparameters $\alpha$ and $\eta$ in addition to the shared MLP model structure sweep. **Position MLP** uses position in the slate as a feature during training time and simply sets it to 0 for fast performance at inference time. **AR Position MLP** is identical to Position MLP with the exception that the position feature is set to each corresponding slate position at inference time (as such it takes into account position biases). **Greedy LSTM** is an LSTM model with fully-connect layers before and after the recurrent middle layers. We tune the hyperparameters corresponding to the number of layers and their respective widths. We use sequences of documents that form slates as the input at training time, and use single examples as inputs with sequence length 1 at inference time, which is similar to scoring documents as if they are in the first position of a slate of size 1. Then we greedily rank the documents based on their prediction scores. **AR LSTM** is identical to Greedy LSTM during training. During inference, however, it selects documents sequentially by first selecting the best document for position 1, then unrolling the LSTM for 2 steps to select the best document for position 2, and so on. This way it takes into account the context of all previous documents and their positions. **Random** selects slates uniformly randomly from the training set.

**Small-scale experiment** ($n = 100, 1000, k = 10$)**:**

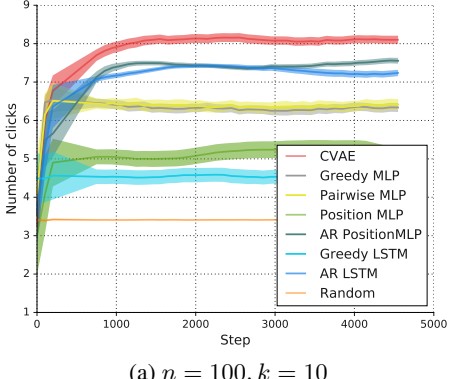
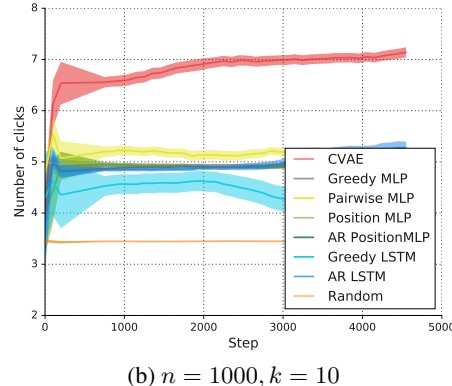

(a) $n = 100, k = 10$                                    (b) $n = 1000, k = 10$

Figure 4: Small-scale experiments. The shaded area represent the 95% confidence interval over 20 independent runs. We compare List-CVAE against all baselines on small-scale synthetic data.

We use the trained document embeddings from the response model for List-CVAE and all the baseline models. For List-CVAE, we also use trained priors $P_\theta(\mathbf{z} \,|\, \mathbf{c}) = \mathcal{N}(\mu^\star, \sigma^\star)$ where $\mu^\star, \sigma^\star = f_{\text{prior}}(\mathbf{c}^\star)$ and $f_{\text{prior}}$ is modeled by a small MLP (16, 32). Additionally, since we found little difference between different hyperparameters, we fixed the width of all hidden layers to 128, the learning rate to $10^{-3}$ and the number of latent dimensions to 16. For all other baseline models, we sweep on the learning rates, model structures and any model specific hyperparameters such as $\alpha, \eta$ for Position MLP and the forget bias for the LSTM model.

Figures 4a and 4b show the performance comparison when the number of documents $n = 100, 1000$ and slate size to $k = 10$. While List-CVAE is not quite capable of reaching a perfect performance of 10 clicks (which is probably even above the optimal upper bound), it easily outperforms all other ranking baselines after only a few training steps. Appendix A includes an additional personalization test.

## 4.2 REAL-WORLD DATA

Due to a lack of publicly available large scale slate datasets, we use the data provided by the RecSys 2015 YOOCHOOSE Challenge (Ben-Shimon et al., 2015). This dataset consists of 9.2M user purchase sessions around 53K products. Each user session contains an ordered list of products on which the user clicked, and whether they decided to buy them. The List-CVAE model can be used on slates with temporal ordering. Thus we form slates of size 5 by taking consecutive clicked products. We then build user responses from whether the user bought them. We remove a portion of slates with no positive responses such that after removal they only account for 50% of the total number of slates. After filtering out products that are rarely bought, we get 375K slates of size 5 and a corpus of 10,000 candidate documents. Figure 5a shows the user response distribution of the training data. Notice that in the response vector, 0 denotes a click without purchase and 1 denotes a purchase. For example, (1,0,0,0,1) means the user clicked on all five products but bought only the first and the last products.

**Medium-scale experiment ($n = 10,000, k = 5$):**

Similarly to the previous section, we train a two-layer response model that officiates as a new semi-synthetic simulation environment. We use the same hyperparameters used previously. Figure 5b shows that List-CVAE outperforms all other baseline models within 500 training steps, which corresponds to having seen less than $10^{-11}\%$ of all possible slates.

**Large-scale experiment ($n = $ 1 million, 2 millions, $k = 5$):**

We synthesize 1,990k documents by adding independent Gaussian noise $\mathcal{N}(\mathbf{0}, 10^{-2} \cdot \mathbf{I})$ to the original 10k documents and label the synthetic documents by predicted responses from the response model. Thus the new pool of candidate documents consists of 10k original documents and 1,990k synthetic ones, totaling 2 million documents. To match each of the $k$ decoder outputs $(x_1, x_2, \ldots, x_k)$ with

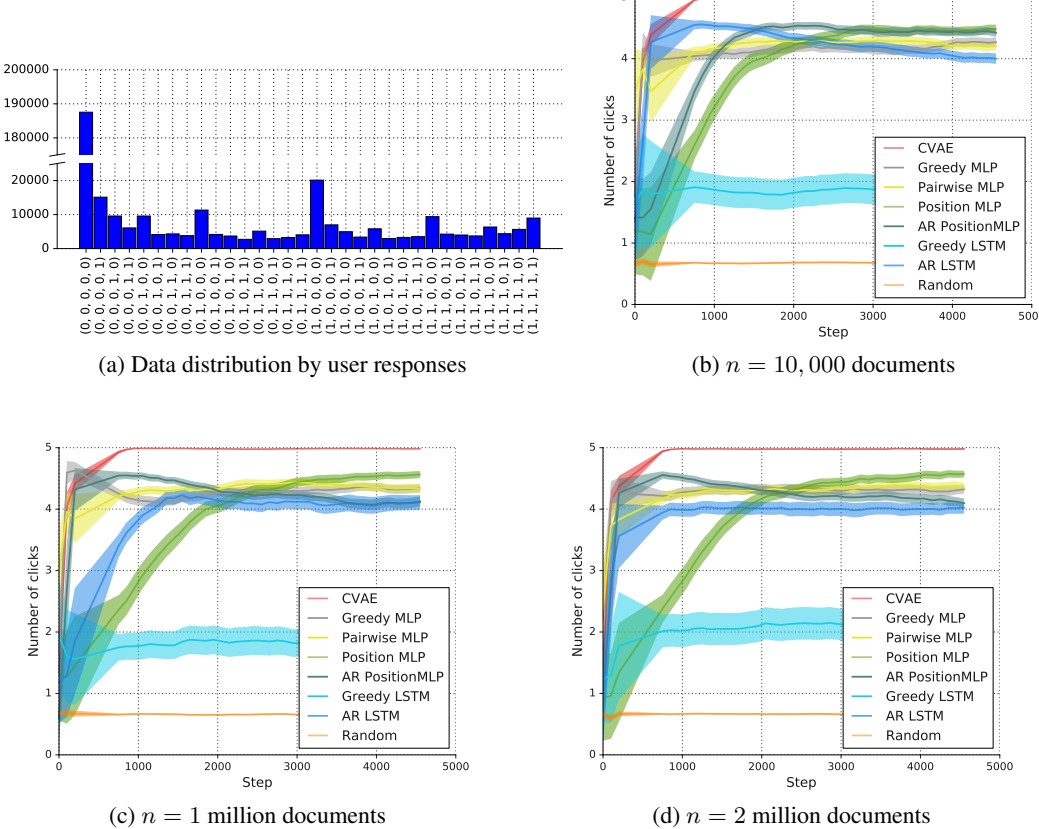

(a) Data distribution by user responses

(b) $n = 10,000$ documents

(c) $n = 1$ million documents

(d) $n = 2$ million documents

Figure 5: Real data experiments: (a) Distribution of user responses in the filtered RecSys 2015 YOOCHOOSE Challenge dataset; (b) We compare List-CVAE against all greedy and auto-regressive ranking baselines as well as the Random baseline on a semi-synthetic dataset of 10,000 documents. The shaded area represent the 95% confidence interval over 20 independent runs; (c), (d) We compare List-CVAE against all baselines on semi-synthetic datasets of 1 million and 2 million documents.

real documents, we uniformly randomly downsample the negative document examples keeping in total only 1000 logits (the dot product outputs in the decoder) during training. At inference time, we pick the argmax for each of $k$ dot products with the full embedding matrix without sampling. This technique speeds up the total training and inference time for 2 million documents to merely 4 minutes on 1 GPU for both the response model (with 40k training steps) and List-CVAE (with 5k training steps). We ran 2 experiments with 1 million and 2 millions document respectively. From the results shown in Figure 5c and 5d, List-CVAE steadily outperforms all other baselines again. The greatly increased number of training examples helped List-CVAE really learn all the interactions between documents and their respective positional biases. The resulting slates were able to receive close to 5 purchases on average due to the limited complexity provided by the response model.

**Generalization test:** In practice, we may not have any close-to-optimal slates in the training data. Hence it is crucial that List-CVAE is able to generalize to unseen optimal conditions. To test its generalization capacity, we use the medium-scale experiment setup on RecSys 2015 dataset and eliminate from the training data all slates where the total user response exceeds some ratio $h$ of the slate size $k$, i.e. $\sum_{i=1}^{k} r_i > hk$ for $h = 80\%, 60\%, 40\%, 20\%$. Figure 6 shows test results on increasingly difficult training sets from which to infer on the optimal slates. Without seeing any optimal slates (Figure 6a) or slates with 4 or 5 total purchases (Figure 6b), List-CVAE can still produce close to optimal slates. Even training on slates with only 0, 1 or 2 total purchases ($h = 40\%$), List-CVAE still surpasses the performance of all greedy baselines within 1000 steps (Figure 6c). Thus demonstrating the strong generalization power of the model. List-CVAE cannot learn much about the

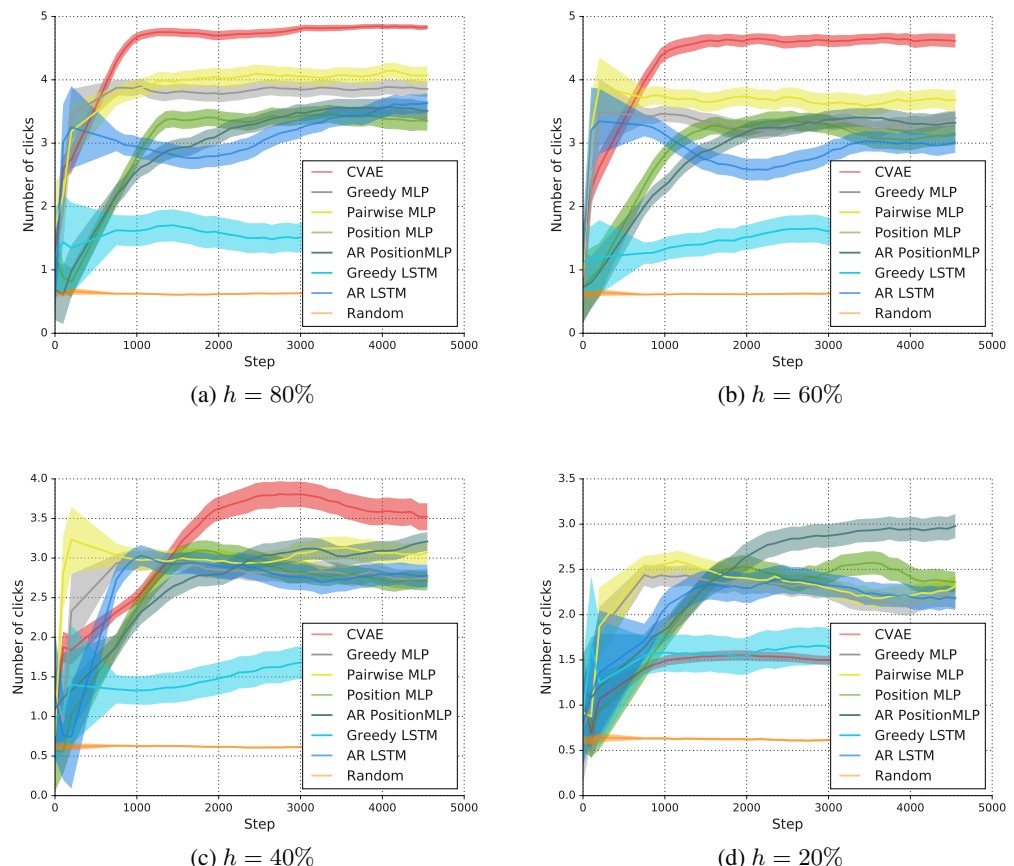

Figure 6: Generalization test on List-CVAE. All training examples have total responses $\sum_{i=1}^{5} r_i \leq 5h$ for $h = 80\%, 60\%, 40\%, 20\%$. Any slates with higher total responses are eliminated from the training data. The other experiment setups are the same as in the Medium-scale experiment.

interactions between documents given only 0 or 1 total purchase per slate (Figure 6d), whereas the MLP-type models learn purchase probabilities of single documents in the same way as in slates with higher responses.

Although evaluation of our model requires choosing the ideal conditioning $\mathbf{c}^{\star}$ at or near the edge of the support of $P(\mathbf{c})$, we can always compromise generalization versus performance by controlling $\mathbf{c}^{\star}$ in practice. Moreover, interactions between documents are governed by similar mechanisms whether they are from the optimal or sub-optimal slates. As the experiment results indicate, List-CVAE can learn these mechanisms from sub-optimal slates and generalize to optimal slates.

## 5 DISCUSSION

The List-CVAE model moves away from the conventional greedy ranking paradigm, and provides the first conditional generative modeling framework that approaches slate recommendation problem using direct slate generation. By modeling the conditional probability distribution of documents in a slate directly, this approach not only automatically picks up the positional and contextual biases between documents at both training and inference time, but also gracefully avoids the problem of combinatorial explosion of possible slates when the candidate set is large. The framework is flexible and can incorporate different types of conditional generative models. In this paper we showed its superior performance over popular greedy and auto-regressive baseline models with a conditional VAE model.

In addition, the List-CVAE model has good scalability. We designed an architecture that uses pre-trained document embeddings combined with a negatively downsampled $k$-head softmax layer that greatly speeds up the training, scaling easily to millions of documents.

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

# A  PERSONALIZATION TEST

This test complements the small-scale experiment. To the 100 documents with slate size 10, we add user features into the conditioning $\mathbf{c}$, by adding a set $\mathcal{U}$ of 50 different users to the simulation engine ($|\mathcal{U}| = 50, n = 100, k = 10$), permuting the innate attractiveness of documents and their interactions matrix $W$ by a user-specific function $\pi_u$. Let

$$r_i^u \sim \mathcal{B} \left[ \left( A_{\pi_u(i)} \prod_{j=1}^{i} W_{i,d_{\pi_u(i)},j,d_{\pi_u(j)}} \right) \Big|_{[0,1]} \right] \tag{7}$$

be the response of the user $u$ on the document $d_i$. During training, the condition $\mathbf{c}$ is a concatenation of 16 dimensional user embeddings $\Theta(u)$ obtained from the response model, and responses $\mathbf{r}$. At inference time, the model conditions on $\mathbf{c}^\star = (\mathbf{r}^\star, \Theta(u))$ for each randomly generated test user $u$. We sweep over hidden layers of 512 or 1024 units in List-CVAE, and all baseline MLP structures. Figure 7 show that slates generated by List-CVAE have on average higher clicks than those produced by the greedy baseline models although its convergence took longer to reach than in the small-scale experiment.

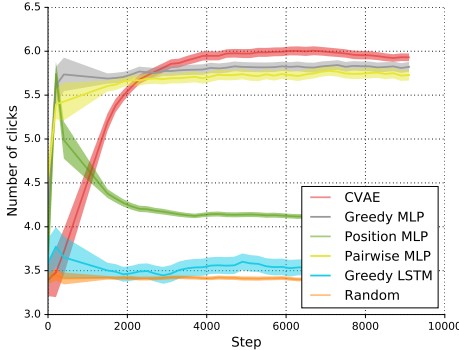

Figure 7: Personalization test with $|\mathcal{U}| = 50, n = 100, k = 10$. The shaded area represent the 95% confidence interval over 20 independent runs. We compare List-CVAE against Greedy MLP, Position MLP, Pairwise MLP, Greedy LSTM and Random baselines on small-scale synthetic data.

