# OpenReview forum: "Beyond Greedy Ranking: Slate Optimization via List-CVAE"
_ICLR.cc/2019/Conference_

### Official Review · AnonReviewer2 · 2018-11-03

**Rating:** 7
**Confidence:** 4

**Review:**

This paper proposes a List Conditional Variational Autoencoder approach for the slate recommendation problem. Particularly, it learns the joint document distribution on the slate conditioned on user responses, and directly generates full slates. The experiments show that the proposed method surpasses greedy ranking approaches.

Pros:
+ nice motivation, and the connection with industrial recommendation systems where candidate nomination and ranker is being used is engaging
+ it provides a conditional generative modeling framework for slate recommendation
+ the simulation experiments very clearly show that the expected number of clicks as obtained by the proposed List-CVAE is much higher compared to the chosen baselines. A similar story is shown for the YOOCHOOSE challenge dataset.

Cons:
- Do the experiments explicitly compare with the nomination & ranking industry standard?
- Comparison with other slate recommendation approaches besides the greedy baselines?
- Comparison with non-slate recommendation models of Figure 1?

Overall, this is a very nicely written paper, and the experiments both in the simulated and real dataset shows the promise of the proposed approach.

---

> ### Author Response · Authors · 2018-11-15
> **Replies to the reviewer's comments [paper updated]**
>
> Thank you for the nice review! We updated the paper to add more baselines to address the reviewer’s comments. The ranking industry standard is generally not well-defined, but in this paper, we mainly compare against the benchmark industry method of the two-stage ranking, which is a weaker performing version of the Greedy MLP baseline due to the candidate generation stage. We also compare List-CVAE with other popular greedy baselines that are suitable for large-scale production settings. For non-greedy baselines, we ran experiments against auto-regressive LSTM (learning contextual/positional biases) and auto-regressive position MLP (learning positional biases) models, which are now included in the results and they performed on par with the greedy baselines.

---

### Official Review · AnonReviewer1 · 2018-11-05
**Using variational auto-encoders to go beyond greedy construction of slates for recommendations**

**Rating:** 6
**Confidence:** 3

**Review:**

The latest revision is a substantially improved version of the paper. The comment about generalization still feels unsatisfying ("our model requires choosing c* in the support of P(c) seen during training") but could spur follow-up work attempting a precise characterization.
I remain wary of using a neural net reward function in the simulated environment, and prefer a direct simulation of Eqn5. With a non-transparent metric, it is much harder to diagnose whether the observed improvement in List-CVAE indeed corresponds to improved user engagement; or whether the slate recommender has gamed the neural reward function. Transparent metrics (that encourage non-greedy scoring) also have user studies showing they correlate with user engagement in some scenarios.
In summary, I think the paper is borderline leaning towards accept -- there is a novel idea here for framing slate recommendation problems in an auto-encoder framework that can spur some follow-up works.
---
The paper proposes using a variational auto-encoder to learn how to map a user context, and a desired user response to a slate of item recommendations. During training, data collected from an existing recommender policy (user contexts, displayed slate, recorded user response) can be used to train the encoder and decoder of the auto-encoder to map from contexts to a latent variable and decode the latent variable to a slate. Once trained, we can invoke the encoder with a new user context and the desired user response, and decode the resulting latent variable to construct an optimal slate.

A basic question for such an approach is: [Fig2] Why do we expect generalization from the user responses c(r) seen in training to c(r*) that we construct for testing? At an extreme, suppose our slate recommendation policy always picks the same k items and never gets a click. We can optimize Eqn3 very well on any dataset collected from our policy; but I don't expect deploying the VAE to production with c(r*) as the desired user response will give us anything meaningful.
The generalization test on RecSys2015-medium (Fig6d) confirms this intuition. Under what conditions can we hope for reliable generalization?

The comment about ranking evaluation metrics being unsuitable (because they favor greedy approaches) needs to be justified. There are several metrics that favor diversity (e.g. BPR, ERR) where a pointwise greedy scoring function will perform very sub-optimally.  Such metrics are more transparent than a neural network trained to predict Eqn6. See comment above for why I don't expect the neural net trained to predict Eqn6 on training data will not necessarily generalize to the testing regime we care about (at the core, finding a slate recommendation policy is asking how best to change the distribution P(s), which introduces a distribution shift between training and test regimes for this neural net).

There are 2 central claims in the paper: that this approach can scale to many more candidate items (and hence, we don't need candidate generation followed by a ranking step); and that this approach can reason about interaction-effects within a slate to go beyond greedy scoring. For the second claim, there are many other approaches that go beyond greedy (one of the most recent is Ai et al, SIGIR2018 https://arxiv.org/pdf/1804.05936.pdf; the references there should point to a long history of beyond-greedy scoring) These approaches should also be benchmarked in the synthetic and semi-synthetic experiments. At a glance, many neural rankers (DSSM-based approaches) use a nearly identical decoder to CVAE (one of the most recent is Zamani et al, CIKM2018 https://dl.acm.org/citation.cfm?id=3271800; the references there should point to many other neural rankers) These approaches should also be benchmarked in the expts.
This way, we have a more representative picture of the gain of CVAE from a more flexible (slate-level) encoder-decoder; and the gain from using item embeddings to achieve scalability.

---

> ### Author Response · Authors · 2018-11-15
> **Replies to the reviewer's comments [paper updated]**
>
> Thank you for the informative review focusing on the following three aspects of the paper, generalization capacity of the model, evaluation metrics and comparison against other baselines. We incorporated the reviewer’s comments into the latest version of the paper, to add a couple of non-greedy baselines and clarify the generalization capacity and the general motivation of our work.
>
> Regarding the generalization ability, given the binary vector format of conditions and the model design, the decoder of the List-CVAE model learns the relationship between the compression of a document and its binary response, and it picks up pairwise (or more generally, sub-slate) interactions from different sub-optimal training slates yet using all of them to construct an optimal slate at test time. We tested the generalization capacity of the model in real world experiments by masking out a percentage of the top responded slates in the Yoochoose dataset at the end of Section 4.2. List-CVAE showed strong generalization capacity and only failed to generalize in the case of training on slates with maximum 1 positive response (h=40%, Figure 6d). This result is expected since List-CVAE can not learn much about the interactions between documents given 0 or 1 positive response, whereas the MLP-type models learn click probabilities of single documents in the same way as in slates with higher responses.
>
> It is true that evaluation of our model requires choosing a c* at or near the edge of the support of P(c). However we can compromise generalization vs. performance by controlling c* to some extent (in this paper, we did not need to use sub-optimal conditioning since the model readily generalizes well to the optimal condition. However in practice, depending on the datasets, one can choose close-to-optimal conditioning at test time for better generalization results). Moreover, interactions between documents are generated by similar mechanisms whether they are from the optimal or sub-optimal slates. Thus the model can learn these mechanisms from sub-optimal slates and generalize to optimal slates (as the experiment results indicate). This discussion has been added to the paper.
>
> On evaluation metrics, it is not always the case that a higher diversity-inclusive score gives better slates measured by user’s total responses. Even though diversity-inclusive metrics are indeed more transparent, they do not directly measure our end goal, which is to maximize the expected number of total responses on the generated slates. We added some clarification on this in the paper.
>
> Regarding our choice of baselines, in this paper, our goal is to challenge the industry state-of-the-art benchmark two-stage ranking on both small and large scales. Therefore we mainly compared List-CVAE with popular baselines that are suitable for large-scale recommender system production settings. For non-greedy baselines, we ran experiments against auto-regressive LSTM (learning contextual/positional biases)  and auto-regressive position MLP (learning positional biases)  models, which are now included in the results and they performed on par with the greedy baselines.
>
> The two models proposed by the reviewer (Ai 2018, Zamani 2018) (we added Ai 2018 as a reference in the paper) are solving information retrieval problems and hence (rightfully) think about slate generation from a ranking paradigm using greedy ranking evaluation metrics such as nDCG, which assumes that “better” documents should be put into higher positions. However, while this is a natural assumption for information retrieval problems, it is the exact assumption that we avoid making since our problem setting is optimal slate generation for maximizing user engagement. One can imagine cases where leaving good quality documents towards the end of the slate encourages users to browse to later positions of the slate, and thus the effects on total user engagement may be diverse.
>
> Finally, we would like to emphasize that we are proposing a paradigm shift for recommender systems that aim to maximize user engagements on whole slates, departing from the traditional viewpoint of a ranking problem and to adopt a direct slate generation framework. As such, we call for new baselines and evaluation metrics that are representative of the new paradigm.

---

> > ### Author Response · Authors · 2018-11-29
> > **A gentle nudge for feedback [paper updated]**
> >
> > Thanks again for your insightful review. We would like to know your thoughts on our rebuttal if possible. We have since updated the paper and would appreciate any additional feedback you have.

---

### Official Review · AnonReviewer3 · 2018-11-06
**Scalable method for the slate-recommendation task**

**Rating:** 6
**Confidence:** 4

**Review:**

This paper pr poses a conditional generative model for slate-based recommendations. The idea of slate recommendation is to model an ordered-list of items instead of modelling each item independently as is usually done (e.g., for computational reasons). This provides a more natural framework for recommending lists of items (vs. recommending the items with the top scores).

To generate slates, the authors propose to learn a mapping from a utility function (value) to an actual slate of products (i.e., the model conditions on the utility).  Once fitted, recommending good slates is then achieved by conditioning on the optimal utility (which is problem dependant) and generating a slate according to that utility. This procedure which is learned in a conditional VAE framework effectively bypasses the intractable combinatorial search problem (i.e., choosing the best ordered list of k-items from the set of all items) by instead estimating a model which generates slates of a particular utility. The results demonstrate empirically that the approach outperforms several baselines.

This idea seems promising and provides an interesting methodological development for recommender systems. Presumably this approach, given the right data, could also learn interesting concepts such as substitution, complementarity, and cannibalization.

The paper is fairly clear although the model is never formally expressed: I would suggest defining it using math and not only a figure.  The study is also interesting although the lack of publicly available datasets limits the extent of it and the strength of the results. Overall, it would be good to compare to a few published baselines even if these were not tailored to this specific problem.


A few detailed comments (in approximate decreasing order of importance):

- Baselines. The current baselines seem to focus on what may be used in industry with a specific focus on efficient methods at test time.

  For this venue, I would suggest that it is necessary to compare to other published baselines. Either baselines that use a similar setup or, at least, strong collaborative filtering baselines that frame the problem as a regression one.

  If prediction time is important then you could also compare your method to others in that respect.

- training/test mismatch. There seems to be a mismatch between the value of the conditioning information at train and at test. How do you know that your fitted model will generalize to this "new" setting?

- In Figures: If I understand correctly the figures (4--6) report test performance as a function of training steps. Is that correct?

  Could you explain why the random baseline seems to do so well? That is, for a large number of items, I would expect that it should get close to zero expected number of clicks.

- Figure 6d. It seems like that subfigure is not discussed. Why does CVAE perform worse on the hardest training set?

- The way you create slates from the yoochoose challenge seems a bit arbitrary. Perhaps I don't know this data well enough but it seems like using the temporal aspects of the observations to define a slate makes the resulting data closer to a subset selection problem than an ordered list.

- Section 3. It's currently titled "Theory" but doesn't seem to contain any theory. Perhaps consider renaming to "Method" or "Model"

---

> ### Author Response · Authors · 2018-11-15
> **Replies to the reviewer's comments [paper updated]**
>
> Thank you for the thoughtful review. We updated the paper to reflect several of the reviewer’s comments, which will be mentioned below. In this paper we mainly compared List-CVAE with popular baselines that are suitable for large-scale production settings. For non-greedy baselines, we ran experiments against auto-regressive LSTM (learning contextual/positional biases) and auto-regressive position MLP (learning positional biases) models, which are now included in the results and they performed on par with the greedy baselines.
>
> In terms of prediction time, all baselines (except auto-regressive methods) and List-CVAE have a run-time complexity of O(k * log(n)) where k is the slate size and n is the number of documents. To obtain logarithmic scaling, one can use kd-trees [Sproull, R.F., Refinements to nearest-neighbor searching in k-dimensional trees. Algorithmica 6(4) (1991) 579–589]. Given this, providing exact numbers becomes highly dependent on the underlying implementation details. With that said, in our experiments, all greedy baselines and the CVAE model are performing below 0.04 ms per test example on a single GPU, and their differences are very small (less than 0.01 ms). The auto-regressive LSTM is ~10 times slower as expected.
>
> Regarding the generalization ability of the model, we performed a test in the Yoochoose dataset by masking out a percentage of the top responses at the end of Section 4.2. List-CVAE only failed to generalize to the case of training on slates with maximum 1 positive response (h=40%, Figure 6d). This result is expected since List-CVAE can not hope to learn much about the interactions between documents given only 0 or 1 positive response per slate, whereas the MLP-type models learn click probabilities of single documents in the same way as in slates with higher responses. This discussion is now added to the paper.
>
> It is true that evaluation of our model requires choosing a c* at or near the edge of the support of P(c). However we can compromise generalization vs performance by controlling c* to some extent (in this paper, we did not need to use sub-optimal conditioning since the model readily generalizes well to the optimal condition. However in practice, depending on the datasets, one can choose close-to-optimal conditioning at test time for better generalization results). Moreover, interactions between documents are generated by similar mechanisms whether they are from the optimal or sub-optimal slates. Thus the model can learn these mechanisms from sub-optimal slates and generalize to optimal slates (as the experiment results indicate).
>
> The figures (4--6) do report test/eval performance as a function of training steps. Due to the setup (Eq. 5) of our simulation environments, a random slate has on average over 3 clicks. On the Yoochoose dataset, in Section 4.2, the paper explained that “we removed slates with no positive responses such that after removal they only account for 50% of the total number of slates” (in order to save training time). Therefore the random slates from the training set (a clarification of this baseline has been added to the paper) have on average slightly over 0.5 purchases.
>
> Given that there are few publicly available slate datasets, we had to devise slates using the temporal order of clicks/purchases within each user session. If the ordering were not important, it would have considerably weakened the performance of List-CVAE.
>
> Other comments (e.g.: Theory -> Method) have been addressed in the newly upload version of the paper.

---

> > ### Comment · AnonReviewer3 · 2018-11-24
> > **Replies to your rebuttal.**
> >
> > 1) You write: "In this paper we mainly compared List-CVAE with popular baselines that are suitable for large-scale production settings."
> >
> > I understand the rationale but given the venue, I think it is necessary not to restrict yourselves to "industry-specific" baselines  (of course you can also discuss computation aspects). I understand that there is some subjectivity in this.
> >
> >
> > 2) You write "Regarding the generalization ability of the model, we performed a test in the Yoochoose dataset by masking out a percentage of the top responses at the end of Section 4.2. List-CVAE only failed to generalize to the case of training on slates with maximum 1 positive response (h=40%, Figure 6d)."
> >
> > That's interesting, thanks for pointing it out. I still have two comments about it:
> >
> > a) Isn't that a more typical setting (i.e., in many domains I would imagine that users would consume a small number of recommended items from a slate)? (This comment also relates to your response about the results reported in Figure 4--6)
> >
> > 2) My original comment was related to training/test mismatch. In addition to showing empirically that it works, it would be nice to provide insights as to why it works in practice with your proposed model. You provide some of that in your rebuttal (in the paragraph that starts with "It is true that evaluation of our model requires choosing a c*") but I am wondering if you have any insights about how the model "generalizes" to different conditioning information?
> >
> > 3) "Given that there are few publicly available slate datasets, we had to devise slates using the temporal order of clicks/purchases within each user session. If the ordering were not important, it would have considerably weakened the performance of List-CVAE."
> >
> > Perhaps I'm missing something here. I am not sure I understand how that would have been detected. In other words, weakened compared to what? One perhaps convincing experiment would be to compare to the performance of List-CVAE trained on the randomly shuffled slates (created in the same way from the same dataset). Of course, I am not suggesting you have to run this experiment but it would perhaps make your argument more convincing.
> >
> > Thanks again for providing feedback and updating your paper.

---

> > > ### Author Response · Authors · 2018-11-29
> > > **Second reply to reviewer's comments**
> > >
> > > 1) "I understand the rationale but given the venue, I think it is necessary not to restrict yourselves to "industry-specific" baselines  (of course you can also discuss computation aspects). I understand that there is some subjectivity in this."
> > >
> > > Auto-regressive baselines (Position and LSTM) have been added to the paper. These methods can, in theory, capture some level of positional and contextual bias. However, we want to point out that these models are computationally more expensive and, as such, are rarely used in time-critical applications. We haven't found any compelling alternative method that can capture contextual bias without a significant overhead in computation.
> > >
> > > 2) a) "Isn't that a more typical setting (i.e., in many domains I would imagine that users would consume a small number of recommended items from a slate)? (This comment also relates to your response about the results reported in Figure 4--6)"
> > >
> > > While in some problems (typically information retrieval settings) most slates have max 1 positive response by design, in the recommendation problems that aim at maximizing user engagement, slates often do have multiple clicks (please refer to papers such as [Katariya16: http://proceedings.mlr.press/v48/katariya16.pdf]). As you correctly pointed out, we haven't run any experiment where slates "naturally" have either 0 or 1 click (the current experiments still target multiple clicks, even if only 0/1 click slate are used during training). Running such experiment will likely require rethinking how the ideal condition is selected, and we do plan to focus on this aspect in future work. Current experiments seem to indicate that we can get away with a slightly overestimated ideal condition.
> > >
> > >
> > > b) "My original comment was related to training/test mismatch. In addition to showing empirically that it works, it would be nice to provide insights as to why it works in practice with your proposed model. You provide some of that in your rebuttal (in the paragraph that starts with "It is true that evaluation of our model requires choosing a c*") but I am wondering if you have any insights about how the model "generalizes" to different conditioning information?"
> > >
> > > Generalization will heavily depend on the underlying dataset. List-CVAE does assume that we can correctly estimate the conditional distribution of slates w.r.t. a given target engagement vector. When some engagement vectors are missing from the training data, List-CVAE's performance degrades (as shown in Figure 6). Prior experiments (on additional synthetic data) and current experiment show that List-CVAE tends to generalize, but we do not have any theoretical insight as to why this is the case.
> > >
> > > 3) "Perhaps I'm missing something here. I am not sure I understand how that would have been detected. In other words, weakened compared to what? One perhaps convincing experiment would be to compare to the performance of List-CVAE trained on the randomly shuffled slates (created in the same way from the same dataset). Of course, I am not suggesting you have to run this experiment but it would perhaps make your argument more convincing."
> > >
> > > The experiment suggested is a good idea: intuitively, we expect that List-CVAE trained on randomly shuffled slates will perform poorly when order is important. In particular, it will be impossible to learn positional bias (since positions will be random). The fact that List-CVAE performs well in our slate version of the Yoochoose dataset seem to indicate that order is important in this dataset - however, we will run the suggested experiment to test this hypothesis.

---

> > > > ### Comment · AnonReviewer3 · 2018-12-04
> > > > **Thank you.**
> > > >
> > > > I read your reply, thank you for the new experiments and detailed description.
> > > >
> > > > Regarding the setting (multiple clicks instead of a single click), I do believe that this is particular even for recommender systems and it's worth saying it explicitly early in the paper.

---

### Meta-Review · Area_Chair1 · 2018-12-19
**novel take on slate generation for recommendation using conditional VAEs**

**Confidence:** 4
**Recommendation:** Accept (Poster)

**Metareview:**

The paper presents a novel perspective on optimizing lists of documents ("slates") in a recommendation setting. The proposed approach builds on progress in variational auto-encoders, and proposes an approach that generates slates of the desired quality, conditioned on user responses.

The paper presents an interesting and promising novel idea that is expected to motivate follow-up work. Conceptually, the proposed model can learn complex relationships between documents and account for these when generating slates. The paper is clearly written. The empirical results show clear improvements over competitive baselines in synthetic and semi-synthetic experiments (real users and clicks, learned user model).

The reviewers and AC also note several potential shortcomings. The reviewers asked for additional baselines that reflect current state of the art approaches, and for comparisons in terms of prediction times. There are also concerns about the model's ability to generalize to (responses on) slates unseen during training, as well as concerns about the realism of the simulated user model in the evaluation. There were questions regarding the presentation, including model details / formalism.

In the rebuttal phase, the authors addressed the above as follows. They added new baselines that reflect sequential document selection (auto-regressive MLP and LSTM) and demonstrate that these perform on par with greedy approaches. They provide details on an experiment to test generalization, showing both when the model succeeds and where it fails - which is valuable for understanding the advantages and limitations of the proposed approach. The authors clarified modeling and evaluation choices.

Through the rebuttal and discussion phase, the reviewers reached consensus on a borderline / lean to accept decision. The AC suggests accepting the paper, based on the innovative approach and potential directions for follow up work.